# Feasibility of Defatted Juice from Sea-Buckthorn Berries (*Hippophae rhamnoides* L.) as a Wheat Beer Enhancer

**DOI:** 10.3390/molecules27123916

**Published:** 2022-06-18

**Authors:** Justyna Belcar, Józef Gorzelany

**Affiliations:** Department of Food and Agriculture Production Engineering, University of Rzeszow, 4 Zelwerowicza Street, 35-601 Rzeszów, Poland; gorzelan@ur.edu.pl

**Keywords:** sea-buckthorn, defatted juice, wheat beer, beer quality, bioactive compounds, antioxidant potential of beer

## Abstract

Juice made from sea-buckthorn berries (*Hippophae rhamnoides* L.) is a valuable source of bioactive compounds, vitamins, as well as micro- and macronutrients. By applying defatted sea-buckthorn juice, it is possible to enhance wheat beer and change its sensory properties and the contents of bioactive compounds in the finished product. A sensory assessment showed that wheat beers with a 5% *v*/*v* addition of sea-buckthorn juice were characterised by a balanced taste and aroma (overall impression). Physicochemical analyses showed that, compared to the control samples, wheat beers enhanced with defatted sea-buckthorn juice at a rate of 5% *v*/*v* or 10% *v*/*v* had high total acidity with respective mean values of 5.30 and 6.88 (0.1 M NaOH/100 mL), energy values lower on average by 4.04% and 8.35%, respective polyphenol contents of 274.1 mg GAE/L and 249.7 mg GAE/L, as well as higher antioxidant activity (measured using DPPH, FRAP, and ABTS assays). The findings show that the samples of wheat beer enhanced with sea-buckthorn juice had average ascorbic acid contents of 2.5 and 4.5 mg/100 mL (in samples with 5% *v*/*v* and 10% *v*/*v* additions, respectively) and contained flavone glycosides, e.g., kaempferol-3-O-glucuronide-7-O-hexoside. Based on the current findings, it can be concluded that wheat beer enhanced with sea-buckthorn juice could emerge as a new trend in the brewing industry.

## 1. Introduction

Beer is a type of beverage which contains four main ingredients: malt, hops, water, and yeast. In wheat beers, some of the barley malt (most commonly from 40 to 60% of the total input material) is replaced with wheat malt or unmalted wheat grain [1]. Wheat beers are characterised by an original flavour owing to the wide range of chemical compounds (e.g., phenols, aldehydes, and esters and their derivatives) produced in the process of top fermentation that contribute to the flavour, which can resemble vanilla, cloves, bananas, or fresh fruit; the effect is produced by the interaction of the two types of malt (barley and wheat) as well as the addition of hops. The final product of the brewing process is characterised by delicate and stable frothy foam, a slightly bitter taste, and haziness [2,3,4]. Wheat beers also have high contents of antioxidant compounds including polyphenols [5].

In recent years, a trend has been observed in consumers’ increasing preference for fruit beers, which mainly include morello cherry, raspberry, banana, and strawberry, as well as other exotic fruit beers. The fruit component may be introduced by adding pulp, juice, concentrate, or aroma, most commonly during the fermentation process. By enhancing beer products with fruit, it is possible to improve their sensory qualities (such as colour, aroma, and taste), and to increase the health-promoting properties of the beverage, which are linked to higher contents of bioactive compounds (e.g., polyphenols), resulting in the higher antioxidant activity of fruit beers compared to traditional beer products [6,7]. In global markets we can encounter the very popular Radler-style beverages, i.e., a combination of beer and flavoured sugar syrup or fruit juice [8,9], as well as Belgian specialty beers produced as a result of spontaneous fermentation with the addition of raspberries or morello cherries, respectively, known as ‘Framboise’ and ‘Kriek’ Lambic beer [6,10,11].

The fruit of sea-buckthorn (*Hippophae rhamnoides* L.), depending on the variety, are yellow to orange in colour, which results from the high concentrations of carotenoids (including lutein, carotene, and zeaxanthin). Sea-buckthorn berries have also been reported to have high contents of health-promoting compounds such as polyphenols (e.g., flavanols and chlorogenic acid), organic acids, micro- and macronutrients, and vitamins [12,13,14,15], including very high levels of ascorbic acid (on average from 53 up to 1550 mg·100 g^−1^ [14,16]), and they do not contain ascorbinase enzyme responsible for the decomposition of ascorbic acid [17]. Owing to their contents of terpenes, alcohols, tannins, and aldehydes, sea-buckthorn berries have a characteristic aroma [12].

The extraction of juice from sea-buckthorn berries is a complex process that may produce changes in the chemical composition and bioavailability of nutrients in the final product. Depending on the preservation process applied (e.g., high-temperature short-term method; HTST), the changes that take place affect the sensory properties, mainly the taste of the juice; on the other hand, a short-term thermal treatment process does not lead to the degradation of ascorbic acid, the content of which decreases during the production of the juice by about 5–11% in relation to the amount of this compound in fresh fruit. Additional technological processes, such as filtration and clarification, contribute to a decrease in the ascorbic acid content. The high-pressure processing (200–600 MPa) that is applied to preserve juice does not produce changes in the quality of the final product; however, it results in a reduced size of particles contained in the juice, enhancing the yellow-orange colour of sea-buckthorn juice [18,19]. Both sea-buckthorn berries and sea-buckthorn juice contain fatty acids, including oleic and palmitoleic acids, phospholipids, and phytosterols [17]. Because these compounds are present, sea-buckthorn juice must be separated into an aqueous fraction and an oil fraction. Lipids present in the oil fraction bind into protein–lipid complexes with soluble low-molecular-weight proteins originating from the malts and they reduce the stability of the structure of the beer head, which comprises carbon dioxide molecules [3,4,20]. The high total acidity of sea-buckthorn juice, which is on average in the range of 2.1–9.1 g·100 mL^−1^ depending on the variety of the raw material, is significantly related to the content of malic and quinic acid (90% share in organic acids; [19]). The bitter and pungent taste of sea-buckthorn juice can be balanced in combination with other beverages, including tea, coffee, wine, and beer [21]. The addition of defatted sea-buckthorn juice to wheat beer, which is characterised by a delicate, slightly sweet taste, may be an interesting and original option acceptable for consumers.

The purpose of this study was to identify the physicochemical, sensory, and antioxidant properties of wheat beers produced with the addition of defatted sea-buckthorn juice. The study also assessed the applicability of the findings to expand the assortment of fruit beers and to make use of sea-buckthorn berries in a new sector of the food industry.

## 2. Results and Discussion

### 2.1. Physicochemical Characteristics of the Wheat Beers

The findings describing the physicochemical parameters of wheat beers enhanced with defatted sea-buckthorn juice are shown in Table 1.

The contents of the apparent extract in the wheat beer samples were in the range of 3.33–4.06% m/m; significantly higher contents of the apparent extract were found in the beer samples with defatted sea-buckthorn juice added at a rate of 10% *v*/*v* (E10 and L10, Table 1). The highest contents of real extract and original extract were identified in the control samples (E0 and L0; Table 1).

The course of the fermentation process and the degree of the final fermentation affect the content of ethyl alcohol, the basic component of beer-type beverages, which is responsible for the sensory characteristics of beer that are perceived by consumers [22]. The degree of the final apparent fermentation in the wheat beer samples ranged from 69.88 to 77.62%, with the highest values identified in the control samples (E0 and L0). An increase in the concentration of defatted sea-buckthorn juice led to a significant decrease in the final apparent fermentation by an average of 3.69% in the samples with a 5% *v*/*v* addition of sea-buckthorn juice and by 8.33% in the samples with sea-buckthorn juice added at a rate of 10% *v*/*v* (Table 1). The values of the final true fermentation identified in the wheat beer samples were less varied but statistically different, independent from the beer samples acquired from wheat malt produced from grains of ‘Lawina’ and ‘Elixer’ wheat varieties. Similar to the degree of the final fermentation, the highest alcohol contents were identified in the control samples (E0 and L0), whereas the beer samples with defatted sea-buckthorn juice added at a rate of 5% *v*/*v* and 10% *v*/*v* were found with ethanol contents that were lower on average by 4.02% and by 9.19%, respectively (Table 1). According to Gasiński et al. [23], fruit beer should have a higher ethyl alcohol content compared to beer that is not enhanced with fruit. The lower contents of ethyl alcohol in the investigated wheat beer samples could be linked to the addition of defatted sea-buckthorn juice, which contains relatively low concentrations of total sugars (on average 4.94–5.72% relative to the variety) and reducing sugars (on average 1.59–1.83% relative to the variety; [13]); these are processed by the yeast in the fermentation process only to a small degree. Furthermore, the addition of sea-buckthorn juice led to an increase in the volume of the finished beer product while decreasing the concentration of the ethanol in the investigated wheat beer samples. A study by Nordini and Garaguso [10] showed that apple beer had an alcohol content of 5.2% *v*/*v*, whereas beer samples enriched with orange peel were found with an ethanol content of 6.0% *v*/*v*. On the other hand, Baigts-Allende et al. [6] reported an alcohol content of 4.0–8.2% *v*/*v* in citrus beer and 2.5–3.5% *v*/*v* in apple beer. Yang et al. [7] investigated apple beer and cranberry beer and reported ethanol contents of 3.5% *v*/*v* and 3.6% *v*/*v*, respectively. Patraşcu et al. [9] reported ethanol contents in lemon beer samples in the range of 1.9–4.0% *v*/*v*, in grapefruit beer samples of 1.9–2.5% *v*/*v*, and in cranberry beer samples of 4.0% *v*/*v*. The high ethanol content of the investigated wheat beers, in particular in the control samples (E0 and L0), corresponded to a relatively high calorific value of the finished product, which ranged from 54.48 to 57.22 kcal/100 mL; on the other hand, the addition of defatted sea-buckthorn juice led to a decrease in the calorific value of the wheat beer samples by an average of 6.2% (Table 1).

Wheat beer as a rule is darker in colour compared to barley beer (depending on the beer style). The wheat beer samples produced without the addition of defatted sea-buckthorn juice (E0 and L0) were found with a slightly darker colour; however, an increased concentration of sea-buckthorn juice added to the beer led to a lighter colour in the wheat beer samples (Figure 1 and Figure 2; Table 1). Baigts-Allende et al. [6] reported that barley beer produced with the addition of citric fruit was found with a colour of 5.8 EBC units, whereas apple beers were characterised by a slightly stronger colour in the range of 6.34–9.81 EBC units. Patraşcu et al. [9] assessed the colour of lemon, grapefruit, and cranberry beers and reported respective values of 6.75–6.83 EBC units, 16.98–17.36 EBC units, and 5.55 EBC units.

The addition of defatted sea-buckthorn juice to wheat beer led to a decrease in the pH value by an average of 13.51% in the samples with a 5% *v*/*v* addition of sea-buckthorn juice and by an average of 18.74% in the samples with a 10% *v*/*v* addition of sea-buckthorn juice relative to the control samples (E0 and L0), which corresponded to a significant increase in the acidity (even twofold in the case of the samples with sea-buckthorn juice added at a rate of 10% *v*/*v*; Table 1). Adadi et al. [24] reported that the pH value and acidity of beer enriched with sea-buckthorn berries amounted to 3.9 and 2.2, respectively. A study by Nordini and Garaguso [10] reported that apple beer samples had a pH of 4.42, whereas beer samples enhanced with orange peel were found with a pH value of 4.86. Patraşcu et al. [9] investigated lemon, grapefruit, and cranberry beers, which were found with an acidity of 4.0–4.64; 4.0–4.4; and 3.76, respectively, whereas the pH values in these types of beer were found to be 2.85–3.09, 3.27–3.49, and 2.91, respectively. A lower pH in beer results in a reduced growth of undesirable microflora, and consequently leads to the greater microbiological stability of the finished beer product [23]. The addition of sea-buckthorn juice, characterised by high acidity and a low pH, on the seventh day during the fermentation process further reduces the risk of microbiological contamination, which is of great importance in the production of fruit beer.

All the wheat beer samples were found with similar contents of carbon dioxide (0.44–0.48%; Table 1). Patraşcu et al. [9] reported contents of carbon dioxide in lemon beer samples in the range of 0.48–0.55%, in grapefruit beer samples of 0.52%, and in cranberry beer samples of 0.55%. The contents of bitter substances in the wheat beer samples enhanced with defatted sea-buckthorn juice were at a similar level (17.5–19.7 IBU; Table 1) and the value increased with a higher addition of sea-buckthorn juice and was significantly greater compared to the wheat beer control samples (E0 and L0; Table 1). The bitter taste in the investigated beer samples originates not only from the basic raw material used in the production of the beer (i.e., hops) but also from the added sea-buckthorn juice (polyphenolic compounds present in juice). The bitter taste and contents of bitter substances in beer are significantly impacted by the variety and dose of the hops applied, the degree of isomerisation of α-acids during the process of boiling the wort with hops, and the reaction of proteins with polyphenols contained in the malt [2,25]. Sea-buckthorn berries contain terpenoids, tannins, as well as aldehydes and alcohols, which are responsible for the bitter taste and characteristic aroma in products enhanced with sea-buckthorn berries [12].

### 2.2. Contents of Bioactive Compounds in Wheat Beers

Ascorbic acid is a chemical compound known for its antioxidant properties. The effects produced by ascorbic acid include the strengthening of defence mechanisms; it is also involved in the synthesis of collagen and the absorption of iron in the human body and it promotes the transcription of mRNA and the treatment of scurvy [18,26]. Sea-buckthorn berries were reported to have high contents of ascorbic acid, ranging from 53–131 mg·100^−1^ g [14] to 114–1550 mg·100 g^−1^ [16], depending on the sea-buckthorn variety and the timing of the harvest; on the other hand, they do not contain the ascorbinase enzyme involved in the decomposition of ascorbic acid in fruit and vegetables. Wheat beer samples enhanced with defatted sea-buckthorn juice had low contents of ascorbic acid, and this compound was not identified in the E0 and L0 beer samples (Table 2). Because of the very rapid decomposition of ascorbic acid, this compound is generally not found in fruit beers (e.g., those with the addition of lemon, grapefruit, black currant, and strawberry), although the fruit added to the beer typically has high contents of ascorbic acid (the respective reported values being 25–53 mg·100 g^−1^; 4–34.4 mg·100 g^−1^; 181–215 mg·100 g^−1^; and 41.2–60 mg·100 g^−1^ [27]). A study by Pimentel et al. [28] used camu-camu fruit (ascorbic acid contents in the range of 2.4–3 g·100 g^−1^ of fruit) to enhance Witbier-type beer. The beer was found with an ascorbic acid content of 15.8 mg·100 mL^−1^.

The contents of polyphenols, bitter substances, vitamins, and melanoidins impact the antioxidant potential of beer [29,30]. The addition of sea-buckthorn juice to wheat beer significantly increased the antioxidant activity of the beer samples assessed using three methods (DPPH·, FRAP, ABTS^+.^), and the increase was associated with a higher concentration of defatted sea-buckthorn juice in the wheat beer samples (Table 2). In comparison, Nordini and Garaguso [10] reported that beer samples produced with the addition of orange peel were found with an antioxidant capacity, assessed using ABTS, of 2.67 mM TE/L, and a reducing capacity, shown by a FRAP assay, of 5.65 mM Fe^2+^/L. They also applied FRAP and ABTS assays to measure the antioxidant capacity of apple beer and reported the respective values of 3.08 mM Fe^2+^/L and 1.62 mM TE/L. The addition of juice from persimmons (kaki) to barley beer led to a decrease in the antioxidant capacity, measured using an ABTS assay, from 6.36 mM TE/L (in a control sample made from 100% barley wort) to 1.65 mM TE/L (in samples made from 25% wort and 75% kaki fruit juice [11]). Deng et al. [31] enhanced beer with omija fruit added during the fermentation process and reported antioxidant activity, measured by DPPH assay, of 1.68 mM TE/L, and a reducing capacity, assessed with FRAP, of 2.4 mM Fe^2+^/L. Portuguese commercial fruit beers flavoured with lemon were reported to have antioxidant capacity in the range of 0.035–0.037 mM TE/L, according to DPPH assay, and a level of 0.008 mM TE/L, according to an ABTS assay [32].

Polyphenolic compounds occurring in beer mainly originate from malt (70–80%) and hops [11]. The way raw materials are prepared (refinement of malt), as well as conditions during the processes of mashing and boiling with hops, significantly affect the total polyphenol contents and the degree of the isomerisation of polyphenols in the finished beer product [25]. Polyphenolic compounds have varied chemical structures, which is associated with their diverse capacity for biological activity, including antioxidant activity [33]. The polyphenols contained in beer significantly affect the sensory perceptions of consumers, such as a sense of thickness, a bitter or sour taste, as well as a fullness of flavour. By adding wheat malt (as a part of the input material) it is possible to increase the total polyphenol content in the finished beer product [34]. The total polyphenol contents in wheat beer samples enhanced with sea-buckthorn juice were on average 14.78% (5% *v*/*v* addition) and 6.45% (10% *v*/*v* addition) higher compared to control samples E0 and L0 (Table 3). Sea-buckthorn berries are found with total polyphenol contents in the range of 128–490 mg ·100 g^−1^ depending on the variety and timing of the harvest [13,18]. The content of polyphenolic compounds is significantly affected by the methods applied during juice production, or more specifically by the process of fruit crushing (breaking cell walls), as well as the thermal processes used to preserve the finished product [23]. Nardini and Garaguso [10] reported total polyphenol contents of 639 mg GAE/L and 399 mg GAE/L, respectively, in beer produced with the addition of orange peel and in apple beer (GAE—equivalent of gallic acid). Gasiński et al. [23] investigated beer with the addition of mangoes and reported slightly lower polyphenol contents in the range of 218.6–267.6 mg GAE/L. The addition of persimmon juice led to a decrease in the total polyphenol contents in the beer samples from 433.32 mg GAE/L (25% juice addition) to 290.34 mg GAE/L (75% juice addition [11]). Portuguese commercial fruit beers flavoured with lemon were found with total polyphenol contents in the range of 240–304 mg GAE/L [32].

Polyphenolic compounds in samples of wheat beer enhanced with sea-buckthorn juice were identified based on an analysis of characteristic spectral data: the mass-to-charge ratio m/z and the maximum absorption of radiation. The characteristics of six polyphenolic compounds that were identified are shown in Table 3. All the identified compounds were flavonols, represented by derivatives of kaempferol and quercetin (in glycoside form). Flavone glycosides are known to have strong antineoplastic and antioxidant properties; they are beneficial for patients with cardiovascular disease and transplants [18]. Kaempferol and quercetin glycosides produce a pungent taste in the mouth, and—to a lesser extent—contribute to a bitter taste, which affects the sensory properties of the finished beer product [18]. The flavonol contents in the control wheat beer samples (E0 and L0) were in the range of 2.18–3.15 mg/L, whereas the addition of defatted sea-buckthorn juice led to an increase in the polyphenol concentrations in the finished beer product by an average of 24.18% and by 48.85% in samples with sea-buckthorn juice added at a rate of 5% *v*/*v* and 10% *v*/*v*, respectively (Table 3). The control wheat beer samples (E0 and L0) were found to contain three compounds, i.e., K-3-*O*-sophoroside, K-3-*O*-rutinoside-7-*O*-glucoside, and K-3-*O*-glucoside-7-*O*-glucoside, which possibly originated from the hops added to the wort during the boiling process; their mean contents were 0.79 mg/L, 0.83 mg/L, and 1.06 mg/L, respectively (Table 3; Figure 3, Figure 4 and Figure 5). The kaempferol-*O*-glucoside contained in hops is extracted even after 30 min of wort boiling (depending on the dose of the wort [35]). The contents of kaempferol in barley beers were on average in the range of 0.10–1.64 mg/L [36,37]. The wheat beer samples enhanced with defatted sea-buckthorn juice were found to contain the compounds Q-3-*O*-rutinoside-7-*O*-glucoside, K-3-*O*-glucoside, and K-3-*O*-glucuronide-7-*O*-glucoside (Figure 3, Figure 4 and Figure 5), which were extracted from sea-buckthorn juice during fermentation and may have been rearranged into more complex glycoside derivatives; notably, the concentration of the latter compound was on average two times higher in the wheat beer samples enhanced with sea-buckthorn juice at a rate of 10% *v*/*v* compared to the samples with a 5% *v*/*v* addition of sea-buckthorn juice (Table 3). A study by Guo et al. [38], investigating the fruit of four sea-buckthorn subvarieties, showed the mean contents of Q-3-*O*-rutinoside and Q-3-*O*-glucoside of 32.9 mg·100 g^−1^ d.w. and 39.7 mg·100 g^−1^ d.w., respectively. On the other hand, Chen et al. [39] reported the following mean contents of these polyphenols: Q-3-*O*-rutinoside–52.0 mg·100 g^−1^ d.w. and Q-3-*O*-glucoside–53.3 mg·100 g^−1^ d.w. (d.w.–dry weight).

### 2.3. Sensory Analysis of Wheat Beers

The sensory qualities of the wheat beer samples enhanced with defatted sea-buckthorn juice determine the specific beer style and contribute to the attractiveness of the beverage for consumers. The results of the sensory assessment of the wheat beers performed by a panel of 11 experts are shown in Table 4 and Figure 6 and Figure 7.

The wheat beer samples enhanced with defatted sea-buckthorn juice at a rate of 5% *v*/*v* were found to have the highest sense of flavour (combination of taste and aroma), bitter taste, and saturation, compared to the other beer samples evaluated. With regard to the control beer samples (E0 and L0), the overall impression evaluated by the expert panel was reflected by a score in the range of 3.86–3.91 on a 5-point scale. The lowest rating was identified in the case of beer samples with a 10% *v*/*v* addition of defatted sea-buckthorn juice (Table 4). Out of all the quality properties assessed, the stability of the beer head in the wheat beer samples received the lowest rating, whether or not sea-buckthorn juice was added in the production process. The taste and aroma of the beer are not only affected by the raw materials used but also by the products of the fermentation process (e.g., aldehydes, phenols, and esters), which impact the taste profile of the beer.

The sensory profile of the investigated wheat beer varied; the control samples (with no addition of sea-buckthorn juice) had a grainy and malty flavour produced by such compounds as maltol and furaneol [40]; they also had a refreshing and sweet taste, characteristic for wheat beers (Figure 6 and Figure 7). The sensory assessment showed that the beer samples enhanced with sea-buckthorn juice had a stronger flavour and more refreshing quality as well as an acidic taste, which was more pronounced in the wheat beer samples with a higher proportional share of the juice added. According to the assessing experts, the control samples E10 and L10 would not be accepted by consumers because of their highly acidic taste. The assessing panel expressed an opinion that, irrespective of the wheat variety used to produce the malt, the beer samples had a balanced taste and aroma profile, whether or not sea-buckthorn juice was added during the production process. The interactions taking place during the fermentation and maturation of beer between esters, sulphur compounds, carbonyls, phenolic compounds, alcohols, and organic acids significantly affect the taste of the produced beer [40]. Beer with a distinct fruity flavour, sweet aftertaste, and pleasant aroma is more favoured and desirable for consumers compared to traditional types of beer [24,41].

## 3. Materials and Methods

### 3.1. Material

Grains of two varieties of winter wheat, i.e., ‘Lawina’ and ‘Elixer’, that were used in the production of wheat beer were obtained from a field experiment conducted in 2021 in the village of Kosina (50°04′17″ N 22°19′46″ E), Podkarpackie Region, Poland. Grain of the winter wheat varieties was harvested after achieving full maturity, and following a resting period, it was used for preparing five-day wheat malts (the methodology of the malting process was described in Belcar et al. [42]). The wheat malt of the ‘Elixer’ variety had the following characteristics: extract potential—86.0% d.m., total protein content—11.6% d.m., content of soluble protein—4.71% d.m., diastatic power—331 WK, and degree of final attenuation—82.4%, whereas the wheat malt of the ‘Lawina’ variety had the following characteristics: extract potential—85.1% d.m., total protein content—11.0% d.m., content of soluble protein—4.42% d.m. (d.m.—dry matter), diastatic power—336 WK, and degree of final attenuation—81.7%.

Materials used in the production of beer samples included commercially available barley malt acquired from Viking Malt company (Strzegom, Poland). The barley malt had the following characteristics: extract potential—80.0% d.m., total protein content—11.4% d.m., content of soluble protein—3.75% d.m. (d.m.—dry matter), diastatic power—324 WK, and degree of final attenuation—82.1%. The wheat and barley malts were refined to the required particle size using a Cemotec disc mill manufactured by FOSS. The input material used in the brewing process comprised commercial barley malt at a rate of 60% and wheat malt at a rate of 40%.

Wheat beer samples were enhanced using defatted juice made from sea-buckthorn berries (after sedimentation) that was produced in 2021 by the Szarłat Company (Cibory Gałeckie, Podlaskie Region, Poland). The defatted juice had the following chemical parameters: fat content—0.04 g/100 g, L-ascorbic acid content—44.45 mg/100 mL, extract content—8.44%, and total acidity—3.36 g/100 mL.

### 3.2. Production of Beer

The production process, based on the infusion method, was carried out in the laboratory of the Department of Agricultural and Food Engineering at the University of Rzeszów. Barley malt with a weight of 3.0 kg and wheat malt with a weight of 2.0 kg were refined and placed in a brew kettle ROYAL RCBM-40N (Expondo; Poland; applied at 80% process efficiency) with 15.0 L of water (3 L of water per each kg of malt). The processes of mashing, boiling with hops, and chilling of beer wort were conducted in line with the methodology described by Gorzelany et al. [8].

All six beer wort samples were found with an extract content of 12.0 °P. The chilled wort samples were poured into 30 L fermentation vessels along with the yeast *Saccharomyces cerevisae* Fermentis Safale US-05 (6 × 10^9^/g), earlier subjected to a dehydration process in line with the manufacturer’s instructions (0.58 g d.m./L of wort). The fermentation process was carried out at 21 °C. After the fermentation process had continued for 7 days, defatted sea-buckthorn juice was added to the beer in specified quantities (0, 5, or 10% relative to wort volume) and then the fermentation process continued for the next 14 days. After 21 days, an aqueous solution of sucrose (0.3%) was added and the beer was poured into bottles for refermentation to achieve an adequate level of carbonation. The beer was then kept at 20 °C. Sensory assessment and physicochemical tests were performed one month after the bottling.

Wheat beer produced using malt obtained from the winter wheat variety ‘Elixer’ and with no addition of defatted sea-buckthorn juice is marked E0, whereas the sample with the 5% *v*/*v* addition of defatted sea-buckthorn juice is marked E5 and the sample with the 10% *v*/*v* addition of sea-buckthorn juice is marked E10. Wheat beer produced using malt obtained from the winter wheat variety ‘Lawina’ and with no addition of defatted sea-buckthorn juice is marked L0, whereas the sample with the 5% *v*/*v* addition of defatted sea-buckthorn juice is marked L5 and the sample with the 10% *v*/*v* addition of sea-buckthorn juice is marked L10. A total of six variants of wheat beer were produced.

### 3.3. Analysis of Beer Quality Indicators

Alcohol contents [% m/m and % *v*/*v*], apparent extract [% m/m], real extract [% m/m], original extract in beer [% m/m], degree of final apparent and real fermentation [%], total acidity [0.1 M NaOH/100 mL], pH, colour [EBC units], carbon dioxide contents [%], contents of bitter substances [IBU units], as well as energy value of beer [kcal/100 mL] were determined following the methodology described by Belcar et al. [34]. The analyses were performed in three replications.

### 3.4. Contents of Bioactive Compounds in Wheat Beers

The total contents of polyphenols [mg GAE/L] determined using the Folin–Ciocalteu method, as well as the polyphenol profile of the beer samples, were measured in compliance with the methodology described by Gorzelany et al. [8].

Determination of polyphenolic compounds [mg/L] was carried out using the UPLC equipped with a binary pump, column and sample manager, photodiode array detector (PDA), tandem quadrupole mass spectrometer (TQD) with an electrospray ionization (ESI) source working in negative mode (Waters, Milford, MA, USA) according to the method of Żurek et al. [43]. Separation was performed using the UPLC BEH C18 column (1.7 µm, 100 mm × 2.1 mm, Waters) at 50 °C at a flow rate of 0.35 mL/min. The injection volume of the samples was 5 µL. The mobile phase consisted of water (solvent A) and 40% acetonitrile in water *v*/*v* (solvent B). The following TQD parameters were used: capillary voltage of 3500 V; con voltage of 30 V; con gas flow 100 L/h; source temperature 120 °C; desolvation temperature 350 °C; and desolvation gas flow rate of 800 L/h. Polyphenolic identification and quantitative analyses were performed based on the mass-to-charge ratio, retention time, specific PDA spectra, fragment ions, and comparison of data obtained with commercial standards and literature findings. The analyses were performed in three replications.

Contents of ascorbic acid [mg/100 mL] in sea-buckthorn berries were assessed in conformity with PN-A-04019:1998 [44]. The analyses were performed in three replications.

### 3.5. Antioxidant Activity in Wheat Beers

Antioxidant capacity of fruit beers (assessed using DPPH [mM TE/L], FRAP [mM Fe^2+^/L, and ABTS [mM TE/L] assays) was measured following the methodology described by Gorzelany et al. [8]. The analyses were performed in three replications.

### 3.6. Sensory Assessment in Beers

Sensory assessment was performed by a panel of 11 experts (4 women and 7 men, aged 30–40 years), in a sensory analysis laboratory in line with the EBC 13.13 method [45]. Beer samples, chilled to a temperature of 10 °C and coded, were served in a random order in transparent plastic cups with a capacity of 250 mL. After each test, the experts were given water to rinse their mouths. Sensory analysis of the beer samples was performed using a 5-point scale, assessing the specific quality characteristics, i.e., aroma (5—very strong, distinctive, and pleasant; 1—imperceptible aroma/ unpleasant smell), taste (5—very good; 1—bad); beer head stability (5—highly stable; 1—unstable), bitter taste (5—weak; 1—very strong), and carbonation (5—high; 1—poor or none). The average score described the general impression (5—excellent; 1—poor) related to the investigated wheat beers. Additionally, evaluation of the beer samples in terms of their taste and aroma applied the sensory profile describing the quality characteristics (malty, fruity, sweet, grainy, strong, full, fresh, phenolic, bitter, and sour) in line with EBC 13.12 [46]. The sensory profile of the fruit beer produced with the addition of defatted sea-buckthorn juice was compared to the control beer (no addition of sea-buckthorn juice).

### 3.7. Statistical Analysis

The results of the fruit beer evaluation are shown as mean values and standard deviations. The statistical analyses of the results were computed using Statistica 13.3 (TIBCO Software Inc., Tulsa, OK, USA). The results related to physicochemical characteristics, polyphenol contents, and antioxidant activity of fruit beer samples were examined using the two-factor completely randomized ANOVA with a significance level of α = 0.05. The mean values were compared using the Tukey HSD test.

## 4. Conclusions

The study, designed to assess the feasibility of defatted sea-buckthorn juice as an enhancer to be used in the production of fruit wheat beers, showed that the most balanced sensory profile (intensity, perceived bitter flavour, as well as fruity taste and aroma) is found in beer samples enhanced with juice at a rate of 5% *v*/*v*. Additionally, these beer samples were shown to have better colour, as well as higher polyphenol contents and antioxidant capacity. Although the addition of defatted sea-buckthorn juice at a rate of 10% *v*/*v* positively affected the health-promoting properties of wheat beer (with a mean content of ascorbic acid of 4.5 mg/100 mL), the sensory properties of this type of beer were not acceptable for the assessing panel, mainly due to its highly acidic taste. The enhancement of wheat beer with defatted sea-buckthorn juice at a rate of 5% *v*/*v* could effectively be applied to expand the assortment of fruit wheat beers on offer.

## Figures and Tables

**Figure 1 molecules-27-03916-f001:**
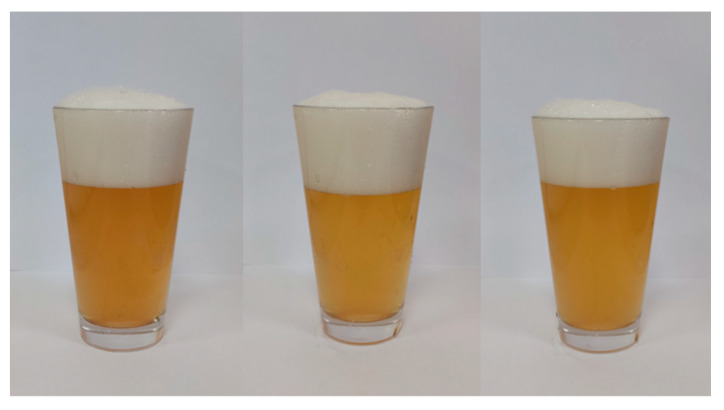
The appearance of the obtained wheat beers (from left)—control (E0); with 5% *v*/*v* addition of defatted sea-buckthorn juice (E5); and with 10% *v*/*v* addition of defatted sea-buckthorn juice (E10).

**Figure 2 molecules-27-03916-f002:**
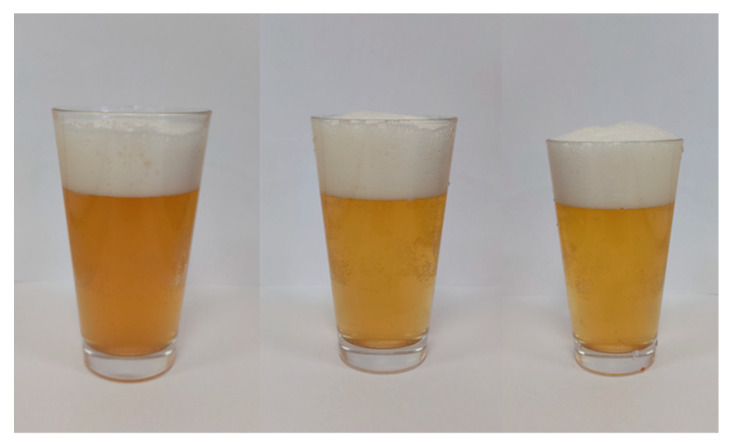
The appearance of the obtained wheat beers (from left)—control (L0); with 5% *v*/*v* addition of defatted sea-buckthorn juice (L5); and with 10% *v*/*v* addition of defatted sea-buckthorn juice (L10).

**Figure 3 molecules-27-03916-f003:**
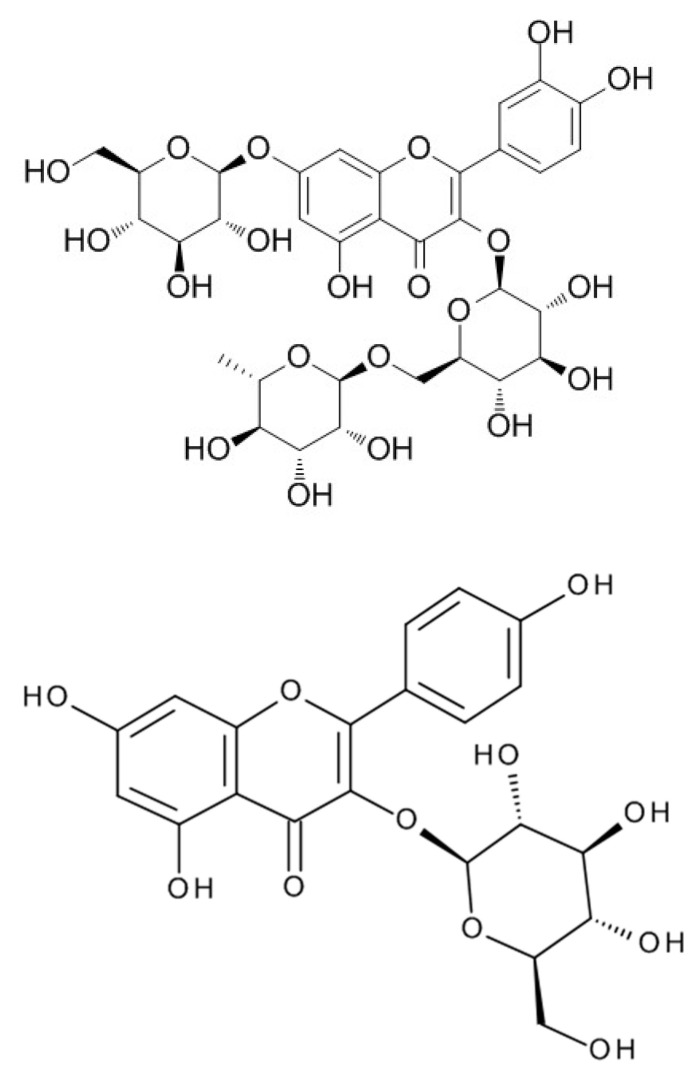
The chemical formula of identified polyphenols—from left—Q-3-*O*-rut-7-*O*-glc and K-3-*O*-glc (source: lgcstandards.com; accessed on 14 June 2022).

**Figure 4 molecules-27-03916-f004:**
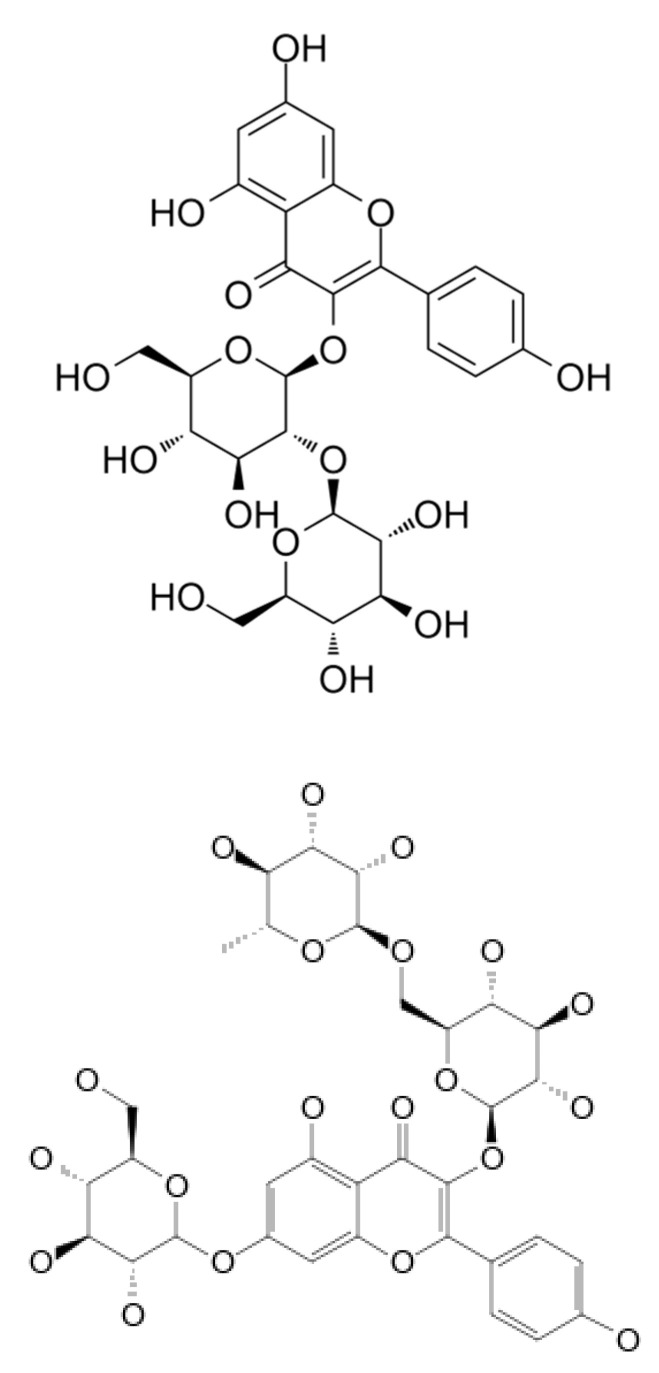
The chemical formula of identified polyphenols—from left—K-3-*O*-sophoroside and K-3-*O*-rut-7-*O*-glc (source: lgcstandards.com; accessed on 14 June 2022).

**Figure 5 molecules-27-03916-f005:**
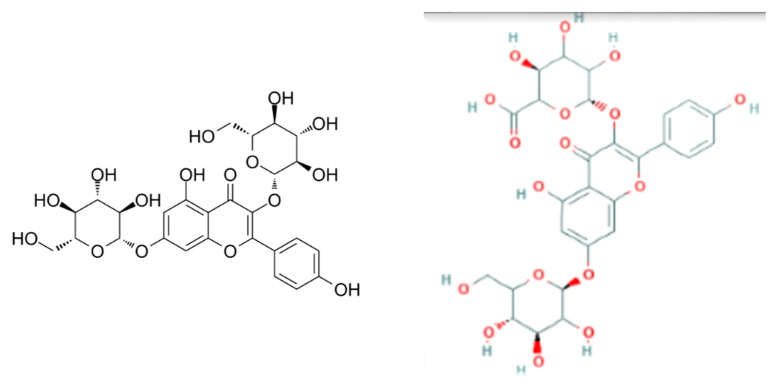
The chemical formula of identified polyphenols—from left—K-3-O-glc-7-O-glc and K-3-O-gluc-7-O-glc (source: lgcstandards.com; accessed on 14 June 2022).

**Figure 6 molecules-27-03916-f006:**
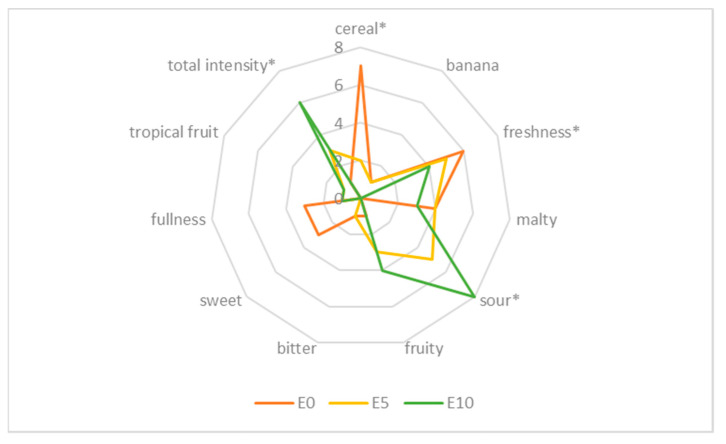
Sensory profile of wheat beer—control sample (E0) and sample with defatted sea-buckthorn juice added at a rate of 5% *v*/*v* (E5) as well as sample with defatted sea-buckthorn juice added at a rate of 10% *v*/*v* (E10). (* marks the attributes that were statistically different at *p* ≤ 0.05).

**Figure 7 molecules-27-03916-f007:**
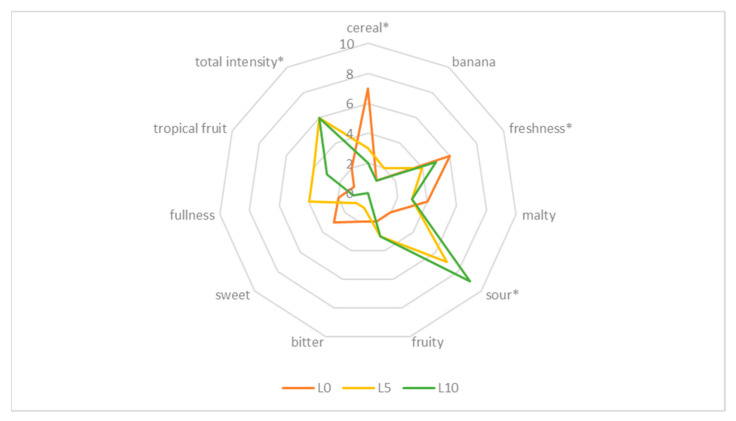
Sensory profile of wheat beer—control sample (L0) and sample with defatted sea-buckthorn juice added at a rate of 5% *v*/*v* (L5) as well as sample with defatted sea-buckthorn juice added at a rate of 10% *v*/*v* (L10). (* marks the attributes that were statistically different at *p* ≤ 0.05).

**Table 1 molecules-27-03916-t001:** Results of physicochemical analysis of wheat beers with defatted sea-buckthorn juice added.

Type of Beer	E0	E5	E10	L0	L5	L10
Apparent extract [%; m/m]	3.33 ^a^ ± 0.06	3.58 ^b^ ± 0.02	4.06 ^d^ ± 0.04	3.52 ^b^ ± 0.02	3.85 ^c^ ± 0.05	4.01 ^d^ ± 0.01
Real extract [%; m/m]	4.81 ^d^ ± 0.03	4.61 ^c^ ± 0.04	4.52 ^b^ ± 0.02	4.51 ^b^ ± 0.01	4.49 ^b^ ± 0.04	4.33 ^a^ ± 0.03
Original extract [%; m/m]	14.88 ^e^ ± 0.10	14.04 ^b^ ± 0.04	13.48 ^a^ ± 0.06	14.38 ^d^ ± 0.08	14.25 ^c^ ± 0.05	13.59 ^a^ ± 0.04
Degree of final apparent attenuation [%]	77.62 ^f^ ± 0.03	74.50 ^d^ ± 0.10	69.88 ^a^ ± 0.02	75.52 ^e^ ± 0.02	72.98 ^c^ ± 0.02	70.49 ^b^ ± 0.01
Degree of final real attenuation [%]	67.67 ^c^ ± 0.05	67.17 ^b^ ± 0.02	66.47 ^a^ ± 0.02	68.64 ^f^ ± 0.04	68.49 ^e^ ± 0.05	68.14 ^d^ ± 0.04
Content of alcohol [%; m/m]	5.28 ^d^ ± 0.05	4.92 ^b^ ± 0.02	4.66 ^a^ ± 0.06	5.16 ^c^ ± 0.06	5.10 ^c^ ± 0.10	4.82 ^b^ ± 0.02
Content of alcohol [%; *v*/*v*]	4.20 ^d^ ± 0.10	3.92 ^b^ ± 0.00	3.71 ^a^ ± 0.01	4.11 ^b^ ± 0.01	4.06 ^b^ ± 0.06	3.84 ^b^ ± 0.02
Colour [EBC units]	25.1 ^d^ ± 0.2	24.1 ^c^ ± 0.1	23.1 ^b^ ± 0.0	25.0 ^d^ ± 0.2	22.9 ^b^ ± 0.3	22.3 ^a^ ± 0.2
Titratable acidity [0.1 M NaOH/100 mL]	3.46 ^b^ ± 0.04	5.44 ^d^ ± 0.03	7.55 ^f^ ± 0.04	3.05 ^a^ ± 0.04	5.15 ^c^ ± 0.05	6.21 ^e^ ± 0.01
pH	4.54 ^c^ ± 0.03	3.95 ^b^ ± 0.05	3.73 ^a^ ± 0.03	4.64 ^d^ ± 0.04	3.99 ^b^ ± 0.05	3.73 ^a^ ± 0.03
Bitter substances [IBU]	15.4 ^b^ ± 0.10	18.1 ^d^ ± 0.10	19.7 ^f^ ± 0.00	14.7 ^a^ ± 0.10	17.5 ^c^ ± 0.00	19.1 ^e^ ± 0.20
Content of carbon dioxide [%]	0.46 ^b^ ± 0.00	0.47 ^b^ ± 0.02	0.44 ^a^ ± 0.02	0.47 ^b^ ± 0.03	0.48 ^b^ ± 0.01	0.44 ^a^ ± 0.02
Energy value [kcal/100 mL]	57.22 ^f^ ± 0.10	53.21 ^c^ ± 0.10	51.02 ^a^ ± 0.06	54.48 ^e^ ± 0.10	53.98 ^d^ ± 0.02	51.35 ^b^ ± 0.10

Data are expressed as mean value (*n* = 3) ± SD; SD—standard deviation. Mean values within a row with different letters are significantly different (*p* < 0.05). E—‘Elixer’ cultivar; L—‘Lawina’ cultivar; 0—wheat beer without defatted sea buckthorn juice; 5—wheat beer with 5% *v*/*v* defatted sea buckthorn juice; 10—wheat beer with 10% *v*/*v* defatted sea buckthorn juice.

**Table 2 molecules-27-03916-t002:** Content of ascorbic acid and antioxidant activity of wheat beers.

Type of beer	E0	E5	E10	L0	L5	L10
Content of ascorbic acid [mg/100 mL]	n.d.	2.5 ^a^ ± 0.4	4.5 ^b^ ± 0.0	n.d.	2.5 ^a^ ± 0.1	4.5 ^b^ ± 0.3
DPPH [mM TE/L]	2.27 ^b^ ± 0.03	2.71 ^d^ ± 0.02	2.76 ^d^ ± 0.06	2.19 ^a^ ± 0.04	2.39 ^c^ ± 0.01	2.74 ^d^ ± 0.04
FRAP [mM Fe^2+^/L]	2.79 ^b^ ± 0.04	3.62 ^d^ ± 0.02	3.89 ^e^ ± 0.07	2.53 ^a^ ± 0.03	3.25 ^c^ ± 0.05	4.09 ^f^ ± 0.04
ABTS^+·^ [mM TE/L]	1.81 ^a^ ± 0.01	2.18 ^c^ ± 0.02	2.47 ^e^ ± 0.01	1.97 ^b^ ± 0.01	2.34 ^d^ ± 0.04	2.62 ^f^ ± 0.02

Data are expressed as mean value (*n* = 3) ± SD; SD—standard deviation. Mean values within a row with different letters are significantly different (*p* < 0.05). E—‘Elixer’ cultivar; L—‘Lawina’ cultivar; 0—wheat beer without defatted sea buckthorn juice; 5—wheat beer with 5% *v*/*v* defatted sea buckthorn juice; 10—wheat beer with 10% *v*/*v* defatted sea buckthorn juice; n.d.—not detected; TE—expressed as Trolox equivalent (mM TE/L).

**Table 3 molecules-27-03916-t003:** Contents of polyphenols and polyphenolic profiles identified by UPLC-PDA-TQD-MS in wheat beer.

					E0	E5	E10	L0	L5	L10
Contents of polyphenols [mg GAE/L]					243.9 ^c^ ± 0.8	277.0 ^f^ ± 0.7	264.6 ^d^ ± 0.6	223.3 ^a^ ± 0.3	271.2 ^e^ ± 0.8	234.8 ^b^ ± 0.5
Compound [mg/L]	Rt [min]	[M-H]^-^ (m/z)	Fragment ions(m/z)	Absorbance maxima (nm)						
Q-3-*O*-rut-7-*O*-glc	3.35	771	609, 301	255, 350	t.c.	0.61 ^b^ ± 0.00	0.55 ^a^ ± 0.05	t.c.	0.53 ^a^ ± 0.05	0.66 ^b^ ± 0.06
K-3-*O*-glc	3.73	447	285	264, 324	t.c.	0.68 ^a^ ± 0.08	0.95 ^b^ ± 0.02	t.c.	0.72 ^a^ ± 0.02	0.90 ^b^ ± 0.15
K-3-*O*-sophoroside	3.97	609	285	264, 324	0.92 ^b^ ± 0.02	t.c.	t.c.	0.65 ^a^ ± 0.06	t.c.	t.c.
K-3-*O*-rut-7-*O*-glc	4.09	755	593, 285	264, 324	0.92 ^b^ ± 0.02	t.c.	t.c.	0.73 ^a^ ± 0.01	t.c.	t.c.
K-3-*O*-glc-7-*O*-glc	4.20	609	447, 285	264, 324	1.31 ^b^ ± 0.04	t.c.	t.c.	0.81 ^a^ ± 0.09	t.c.	t.c.
K-3-*O*-gluc-7-*O*-glc	4.61	623	447, 285	264, 324	t.c.	1.39 ^a^ ± 0.02	2.41 ^b^ ± 0.07	t.c.	1.34 ^a^ ± 0.04	2.20 ^b^ ± 0.22
K-3-*O*-gluc-7-*O*-glc	5.37	623	447, 285	264, 324	t.c.	0.90 ^a^ ± 0.03	1.42 ^c^ ± 0.06	t.c.	0.88 ^a^ ± 0.04	1.34 ^b^ ± 0.03
Total					3.15 ^b^ ± 0.08	3.57 ^c^ ± 0.07	5.33 ^c^ ± 0.10	2.18 ^a^ ± 0.14	3.46 ^c^ ± 0.04	5.09 ^c^ ± 0.34

Data are expressed as mean value (*n* = 15) ± SD; SD—standard deviation. Mean values within a row with different letters are significantly different (*p* < 0.05). E—‘Elixer’ cultivar; L—‘Lawina’ cultivar; 0—wheat beer without defatted sea buckthorn juice; 5—wheat beer with 5% *v*/*v* defatted sea buckthorn juice; 10—wheat beer with 10% *v*/*v* defatted sea buckthorn juice. Q-quercetin; K—kaempferol; glc—glucoside; rut—rutinoside; gluc—glucuronide t.c.—trace content below LOQ; GAE—equivalent of gallic acid (mg GAE/L).

**Table 4 molecules-27-03916-t004:** Sensory analysis of wheat beer.

	E0	E5	E10	L0	L5	L10
Aroma	4.23 ^ab^ ± 0.41	4.18 ^ab^ ± 0.25	3.82 ^a^ ± 0.60	3.95 ^a^ ± 0.57	4.54 ^b^ ± 0.68	4.27 ^ab^ ± 0.46
Taste	3.73 ^a^ ± 0.24	4.27 ^ab^ ± 0.34	3.73 ^a^ ± 0.28	3.91 ^ab^ ± 0.44	4.54 ^b^ ± 0.52	3.77 ^a^ ± 0.28
Foam stability	3.55 ^a^ ± 0.13	3.82 ^a^ ± 0.30	3.41 ^a^ ± 0.17	3.55 ^a^ ± 0.33	3.59 ^a^ ± 0.26	3.50 ^a^ ± 0.44
Bitterness	4.00 ^a^ ± 0.17	4.09 ^a^ ± 0.14	3.82 ^a^ ± 0.25	4.18 ^a^ ± 0.37	4.27 ^a^ ± 0.34	3.64 ^a^ ± 0.12
Saturation	3.73 ^a^ ± 0.34	4.00 ^a^ ± 0.17	3.64 ^a^ ± 0.27	4.00 ^a^ ± 0.29	4.00 ^a^ ± 0.33	3.45 ^a^ ± 0.38
Overall impression	3.86 ^ab^ ± 0.47	4.13 ^ab^ ± 0.37	3.72 ^a^ ± 0.36	3.91 ^ab^ ± 0.66	4.28 ^b^ ± 0.45	3.78 ^a^ ± 0.44

Data are expressed as mean value (*n* = 11) ± SD; SD—standard deviation. Mean values within a row with different letters are significantly different (*p* < 0.05). E—‘Elixer’ variety; L—‘Lawina’ variety; 0—wheat beer without defatted sea buckthorn juice; 5—wheat beer with 5% *v*/*v* defatted sea buckthorn juice; 10—wheat beer with 10% *v*/*v* defatted sea buckthorn juice.

## Data Availability

Not applicable.

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
