# Peer review of "Feasibility of Defatted Juice from Sea-Buckthorn Berries (Hippophae rhamnoides L.) as a Wheat Beer Enhancer"

_molecules, 2022, doi:10.3390/molecules27123916_

Round 1

Reviewer 1 Report

In this manuscript the physicochemical, sensory and antioxidant properties of wheat beers produced with addition of defatted sea-buckthorn juice were studied.

The paper is original and provides a new contribution to the studied field.

The objective of the work is clearly stated and the results well presented and interpreted.

Tables and figures are informative and essentials.

The references cited are relevant, quite updated and in appropriate number.

In my opinion the paper is suitable for publication.

Before acceptance, however, few amendments are necessary.

Tables 1, 2 and 3 are difficult to read. There is a need to improve formatting and better separate columns.

Several acronyms have not been explained. Please explain the first time they appear in the text and in the caption of the tables and figures where they are reported, i.e. TE, GAE, d.m., d.w.

In the manuscript "liter" is sometimes indicated with "L" and sometimes with "l". Please standardize, always "L".

At row 349 change “The” to “the”

At row 399 change “whet” to “wheat”

Author Response

The Authors are grateful for the contribution of the Reviewer.

  1. Tables 1, 2 and 3 are difficult to read. There is a need to improve formatting and better separate columns.

Answer: 

Columns 1 and 2 have been extended and column 3 has been placed horizontally on a separate page.

2. Several acronyms have not been explained. Please explain the first time they appear in the text and in the caption of the tables and figures where they are reported, i.e. TE, GAE, d.m., d.w.

Answer;

The abbreviations are explained in the manuscript.

3. 

In the manuscript "liter" is sometimes indicated with "L" and sometimes with "l". Please standardize, always "L".

At row 349 change “The” to “the”

At row 399 change “whet” to “wheat”

Answer:

The errors have been fixed in manuscript.

Reviewer 2 Report

The Authors evaluated the opportunity to utilize the defatted juice from sea-buckthorn berries as a wheat beer enhancer. They analyze the physico-chemical traits, nutraceutical properties and sensorial features on fruit beer (with 5 and 10% defatted juice from sea-buckthorn berries) of compared to control beer using two varieties of winter wheat i.e., ‘Lawina’ and ‘Elixer’.

Generally, the paper contains some interesting data, and, thus, yields new information in this research area. Introduction is well-structured. This paper is a well-planned and well-written and all aspects of research have been addressed appropriately.

However, the paper should be revised as recommended, and some issues should be changed (see the suggestions for the Authors in attached file).

Author Response

The authors are grateful for the contribution of the Reviewer.

  1. All tables has been modified in manuscript.
  2. In text have been used the same term.
  3. Remarks throughout the text have been corrected.
  4. in material and methods have been added units of measurement.
  5. K-3-O-gluc-7-O-hex it is a compound that had two retention times. therefore it was listed twice in the table.
  6. English language have been checked by native speaker.

Reviewer 3 Report

In the manuscript, the preparation of wheat beer enriched with juice from sea-buckthorn berries is described. The authors analyze the physicochemical properties of beer, present the antioxidant activity of a new beer and analyze the content of polyphenols. Finally, the sensory analysis helps to conclude which of the tested composition of beer is the most acceptable for consumers.

The manuscript is very well written. The introduction helps to understand the context. The results are presented clearly and in detail.      

For better understanding, the chemical formula of identified polyphenols (Table 3) should be presented.

The units TE/L and GAE/L should be defined.   

Author Response

The authors are grateful for the contribution of the Reviewer.

  1. For better understanding, the chemical formula of identified polyphenols (Table 3) should be presented.

Answer:

The chemical formulas of identified polyphenols was added in manuscript.

2. The units TE/L and GAE/L should be defined.   

Answer:

Defines has been added